# Competition Anxiety in Combat Sports and the Importance of Mental Toughness

**DOI:** 10.3390/bs13090713

**Published:** 2023-08-28

**Authors:** Dara Mojtahedi, Neil Dagnall, Andrew Denovan, Peter Clough, Stephen Dewhurst, Matthew Hillier, Kostas Papageorgiou, John Perry

**Affiliations:** 1Department of Psychology, School of Education and Psychology, University of Bolton, Bolton BL3 5AB, UK; 2Department of Psychology, Faculty of Health, Psychology and Social Care, Manchester Metropolitan University, Manchester M15 6BH, UK; 3Department of Psychology, School of Human and Health Sciences, University of Huddersfield, Kirklees HD1 3DH, UK; 4Department of Psychology, School of Psychology and Social Work, Faculty of Health Sciences, University of Hull, Hull HU6 7RX, UK; s.dewhurst@hull.ac.uk; 5Independent Researcher, Huddersfield HD2 2UY, UK; m.hillier9191@gmail.com; 6School of Psychology, Faculty of Engineering and Physical Sciences, Queen’s University Belfast, Belfast BT9 5BN, UK; 7Department of Physical Education and Sports Science, Faculty of Education and Health Sciences, University of Limerick, V94 T9PX Limerick, Ireland; john.l.perry@ul.ie

**Keywords:** mental toughness, competitive anxiety, sportspersonship, sportsmanship, combat sports, martial arts

## Abstract

Combat sports require participants to engage in potentially dangerous forms of contact-based competition. Pressure to succeed, coupled with the risk of severe injury can induce significant levels of anxiety, which if uncontrolled, can negatively impact performance and possibly promote unsporting conduct. The present study examined competitive anxiety levels of combat sports athletes and determined whether self-reported scores were associated with mental toughness and Sportspersonship attitudes. A cross-sectional survey design was used whereby participants (*N* = 194) completed a battery of questionnaires measuring competitive combat sport experiences, demographic details, Sportspersonship traits (compliance towards rules, respect for opponents, and game perspective), and competition anxiety (somatic, cognitive, and self-confidence; reported retrospectively). Results suggest that mentally tough athletes experience lower levels of cognitive and somatic anxiety, and higher self-confidence, prior to competitions. Findings also found that athletes endorsing more altruistic and respectful attitudes in sport (Sportspersonship) reported higher levels of competition anxiety. The findings demonstrate that mental toughness is allied to positive attributes and could potentially be operationalized to improve both the retention and performance of combat sports athletes. Thus, the authors advocate the use of mental toughness coaching interventions within combat sports.

## 1. Introduction

This paper examined the extent to which mental toughness predicted competitive anxiety in combat sports (CS), a category of sports involving physical combat [1]. Prominent examples include boxing, kickboxing, muay thai, taekwondo, mixed martial arts (MMA), and various forms of grappling (e.g., wrestling, Brazilian jiu jitsu, and judo). Depending on the form of combat, competitors attack their opponents using strikes, throws, chokes, joint manipulation, and/or pins [2,3]; due to the fundamental role of violence, CS is considered one of the most dangerous forms of competition [4]. Commensurate with this perspective, approximately 40% of MMA contests conclude as a consequence of injury [5,6]. Acknowledging the nature of CS, most athletes are aware of the inherent risks; for instance, Jensen et al. [3] reported that competitors frequently experienced fear and apprehension prior to contests. This in part centered on the perception that opponents intended to cause them harm. Furthermore, evidence across various sports suggests that fear of physical harm and injury can manifest as competitive anxiety (CA) [7,8].

### 1.1. Competitive Anxiety

CA is a negative emotional response to competition stressors exhibited prior to and during athletic performance [9]. An aversive response occurs when a performance-related situation is perceived as threatening [10,11]. CA is characterized by cognitive (e.g., negative concerns about performance) and somatic (e.g., tremor of limbs) symptoms [12,13]. Although minor levels of CA are typical and somewhat of an expected emotional response during competitions [14,15], higher levels can negatively affect performance [16,17,18,19,20,21,22]. The potential importance of CA is demonstrated by the fact that researchers have been able to, with 62–79% accuracy, use CA levels to predict winners in karate and taekwondo contests, with success being predicted by low somatic and cognitive anxiety, and high self-confidence [23,24]. Pertinent to the present study, Jensen et al. [3] reported that the performance of MMA athletes suffered if they were not able to lower their anxiety and fear to manageable levels prior to the commencement of fighting. The deteriorating effects of CA on performance can be attributed to its various symptoms. In relation to somatic anxiety symptoms, anxiety has been linked to disrupted cognitive–motor performance [25] and increased release of cortisol, a stress hormone that has been associated with poorer physical performance and recovery [26,27]. The impact of cognitive anxiety on performance can be explained using the reflective impulse model [28,29], which posits that task performance is carried out through an intuitive system that is formed by accumulated experience and a reflective system, reliant on the processing capacity in working memory [30]. Though physical actions are intuitive, many of these automatic skills can become consciously regulated and disrupted due to depleted working memory when an individual exhibits extreme anxiety [31]. As such, athletes experiencing cognitive anxiety will become more conscious of their performance and less able to make effective decisions. However, not all athletes react to competition stressors similarly, the Individual Zones of Optimal Functioning (IZOF [32,33]) posits that individual differences exist in the way individuals react to anxiety and other pre-competition emotions with some athletes gaining optimal performance from low levels of arousal, whilst others perform better with moderate or high levels of arousal. Despite some athletes benefiting from competition anxiety, research shows that anxiety tends to have significant negative effects on performance for most athletes (for meta-analysis, see [34]).

In addition to performance-based outcomes, excessive CA can also have a negative impact on other sport-related factors, such as motivation [35,36], confidence [37], enjoyment [38,39,40], perceived risk of injury (see [41,42]), and sport discontinuation [38]. Evidence also specifies that CA could be linked with the endorsement of unsporting attitudes. Kalkan and Yücel [43] observed a negative relationship between state anxiety and Sportspersonship (i.e., respect for the rules, rituals, and traditions of sport [44]), suggesting that CA could provoke individuals to disregard the rules of the sport and/or respect for their opponent during competitions. This would be especially problematic within CA as unsporting attitudes could result in greater infliction of harm and possible disqualification; however, this relationship has only been explored within handball, thus further investigation is needed to determine whether CA promotes unsporting behavior in CS.

### 1.2. Strategies for Coping with Competitive Anxiety

Noting the adverse consequences, sports psychologists have developed strategies to help sportspersons with moderate to high levels of CA. In this context, psychological models of anxiety advise that appropriate appraisal of stressors alleviates physiological arousal and facilitates recovery from stressful events [45,46,47]. This is important since Gould et al. [39] identified a negative interpretation of arousal as an important precursor of CA, thus positively reframing challenging situations could reduce performance-related anxiety. Furthermore, interventions promoting positive restructuring of competition stressors have been found to reduce CA [48]

Self-confidence also plays an important role in the appraisal of competitive stressors—explicitly, the multidimensional theory of competitive state anxiety (MTCSA [49]) proposes that self-confidence influences the perception of outcomes. Hence, individuals with higher levels of self-confidence exhibit lower levels of somatic and cognitive anxiety and can control their negative emotions more effectively [50]. The relationship between self-confidence and positive competitive thinking has led several researchers to identify the attribute as one of the most important variables associated with sports performance [18,51,52].

### 1.3. Mental Toughness

Both self-confidence and adaptive coping (positive appraisal of stressors) are pertinent to Mental Toughness (MT), a broader construct that encompasses the ability to control emotions, positively appraise challenging situations, and maintain confidence under pressure [53]. The term serves as a hypernym for various attributes that facilitate resilience, confidence, and success across a range of achievement contexts [54]. MT was initially applied within the domain of sport psychology to signify a battery of experientially developed and heritable psychological resources (i.e., values, attitudes, emotions, cognitions, and behaviors) that enabled success in sports and physical activity [55].

One frequently used conceptualization of MT is the *4Cs* model proposed by Clough et al. [53]. This includes interrelated, but independent components: *Control* (the degree to which one feels in control of their life and emotions), *Commitment* (the ability to effectively set goals and reach targets consistently), *Challenge* (the tendency to be driven by challenging oneself and perceiving threatening and adverse situations as opportunities for development), and *Confidence* (one’s belief in their own abilities to perform productively and succeed in both competitive and social settings).

MT has been extensively studied by sports psychologists due to its pronounced relationship with sporting success and resilience in general [56]. According to Cowden’s [57] literature review of MT in sports, 90% of studies found that mentally tough athletes performed better than counterparts with low MT. More specific to the present study, Slimani et al. [58] were able to discriminate winners from losers in kickboxing bouts based off of their MT scores. Indeed, MT has been emphasized as a necessary pre-requisite for CS athletes by intercollegiate wrestling coaches and elite kickboxers [59,60]. The positive effects of MT on sporting success can be primarily attributed to a greater commitment to training and motivation to push oneself. In addition to this, MT may also facilitate sporting success by reducing competition anxiety and its adverse effects on performance and commitment. Recent research identified a negative relationship between MT and CA among school athletes [61], due to mentally tough individuals’ abilities to adapt better to stressful situations and perceive threatening situations as being less stressful [62,63,64].

MT attributes such as dealing with pressure [65] and maintaining confidence during competitions [66] have been identified as essential resources for CS success. Promisingly, a growing body of research demonstrates that MT can be fostered through coaching interventions (see [55] for review). It is likely that such skills will also act as protective factors against CA, allowing CS athletes to focus better on the contest, however, the relationship between MT and CA in CS is currently unexplored.

### 1.4. The Present Study

Noting the potential benefits of MT to sporting performance generally, and that mentally tough individuals are more resilient within stressful situations [62,63,64], the present study examined whether MT could predict somatic and cognitive anxiety, as well as self-confidence (i.e., confidence in ability to succeed in a sporting competition). It was hypothesized that MT would be negatively correlated with cognitive and somatic anxiety but positively correlated with self-confidence. Additionally, because unsporting behavior is often used in CS competitions to gain an advantage over an opponent (e.g., provoking an opponent, ignoring the rules), the authors also investigated whether poor Sportspersonship attitudes were related to CA, as alluded in past research (see [43]).

## 2. Materials and Method

### 2.1. Respondents

CS athletes were recruited through advertisements on online combat sport internet pages and direct contact with CS athletes within the UK. Following Tabachnick and Fidell’s [67] formula for regression model sampling [*N* > 50 + 8*m*] (*M* = *N* of predictors included in regression model), a minimum requirement of 82 respondents was calculated. One-hundred and ninety seven participants responded to the online survey, however, three participants were removed due to not having competed in CS. Competitive experience was defined as involvement in a martial arts match that was competitive (i.e., a winner could be declared). The final sample consisted of 194 athletes (180 men; *M*_age_ = 27.8, *SD_age_* = 6.77) who had contested in various CS. The most common primary sport (i.e., main sport respondents competed in) within the sample was MMA (*n* = 81), followed by boxing/kickboxing (including Muay Thai and K1; *n* = 59), grappling (Brazilian Jiu jitsu, wrestling, and judo; *n* = 53), and Karate (*n* = 1).

Due to some CS not containing a clear differentiation between amateur and professional competition, participant competition level was only recorded for participants who had competed in boxing, kickboxing/Muay Thai, or MMA. Within these sports, competition level is differentiated based on rulesets and monetary payment. Using this criteria, the sample was comprised of 85 respondents at amateur level and 80 of whom competed at professional level in boxing/kickboxing or MMA.

Duration of training in primary CS ranged from 1–34 years (*Median* = 7, *IQR* = 6), total CS experience ranged between 1–46 years (*Median* = 9, *IQR* = 10) and total competition experience ranged from 1–40 years (*Median* = 6, *IQR* = 8). However, these variables may not accurately index CS experience due to failing to consider the intensity and consistency of respondents’ competition and training history or distinguish between childhood and adult-level experience. Therefore, these variables were not included as predictors of CS in subsequent analyses.

### 2.2. Materials

#### 2.2.1. Mental Toughness (MT)

MT was measured using the Mental Toughness Questionnaire-10 (MTQ-10 [54]), a shortened version of the Mental Toughness Questionnaire-48 (MTQ-48 [52]). The MTQ-10 has been empirically supported as a unidimensional measure of MT with robust psychometric properties (Comparative fit index > 0.91, [54,68]). The shortened scale was selected for the present study to reduce survey completion time. The ten items describe personal characteristics relating to Clough et al.’s [52] four constructs of MT: *Confidence*, *Control*, *Commitment*, and *Challenge*. Participants were required to report their agreement to each statement (e.g., “I generally feel in control”) using a five-point Likert-scale (1 = Strongly Disagree, 5 = Strongly Agree) and scores were summed to calculate MT (four items are reverse scored). The unidimensional measure of MT demonstrated good internal consistency in the present data (Cronbach a = 0.73, see Table 1).

#### 2.2.2. Competitive Anxiety (CA)

CA was measured using the revised Competitive State Anxiety Inventory-2 (RCSAI-2 [69]). This 17-item scale was compiled by Cox and colleagues in an attempt to validate the Competitive State Anxiety Inventory-2 [49], which was methodologically limited due to an arbitrary item-inclusion process and lack of confirmatory factor analysis validation. The RCSAI-2 assesses CA through three constructs: somatic anxiety (seven items, e.g., “I feel jittery”), cognitive anxiety (five items; “I am concerned about losing”), and self-confidence (five items, e.g., “I am confident about performing well”).

The scale was designed to be completed by participants prior to a competition, reflecting athletes’ current states; however, data for the present study was collected towards the end of the COVID-19 pandemic when sporting competitions were not permitted. Therefore, the present study measured competitive anxiety retrospectively. Participants reported how much each statement reflected their state prior to their last primary CS competition using a four-point Likert-scale (1 = Not At All, 4 = Very Much So). Items for each construct were averaged and multiplied by 10, with higher scores reflecting greater CA.

Due to collecting CA scores retrospectively, it is acknowledged that participants’ CA responses are more indicative of athletes’ reflections of CA rather than their actual states. However, this is unlikely to impact the accuracy of the results, given that research has shown individuals can recall the frequency of CA symptoms experienced from past competitions accurately [70]. All three subscales demonstrated good internal consistency (Cronbach a = 0.84–0.9, see Table 1).

#### 2.2.3. Sportspersonship

Sportspersonship was measured using the Compliant and Principled Sportspersonship Scale (CAPSS [71]). The 24-item scale comprises five subscales, that relate to an athlete’s compliance towards officials (five items, e.g., “I never argue with a referring decision even if I feel it is wrong”), rules of the sport (five items, e.g., “I never break the rules of my sport”), (not) legitimizing injurious acts (four items, e.g., “I would not intentionally injure an opponent to gain advantage”), respect for opponents (four items, e.g., “I will always congratulate my opponent on his or her victory”), and game perspective (six items, e.g., “I do not believe in winning at all costs”). Items were scored using a 4-point Likert scale (1 = Strongly Disagree; 4 = Strongly Agree) and averaged for each subscale. Two of the subscales were removed from the current survey. Compliance towards officials was omitted because the items referred to behaviors that would typically occur after a bout (e.g., arguing with the referee over the decision). Items relating to legitimizing injurious acts were also removed because injurious acts encompass the fundamental purpose of CS, therefore, high scores on the scale would not reflect unsportsmanlike behavior. All three Sportspersonship subscales demonstrated good internal consistency (Cronbach a = 0.83–0.88, see Table 1).

### 2.3. Procedures

Following ethical approval from the lead researcher’s institutional ethics committee, an online survey was published using an online survey platform (*Qualtrics*). Participants were presented with an information sheet that reaffirmed the inclusion criteria and contained a consent form. After providing informed consent, participants were required to complete a series of questions about demographics and combat sport experiences, and then work through the test measures (MT, Sportspersonship, and CA).

## 3. Results

Statistical analyses used SPSS^®^ 26.0 (IBM Corporation, Armonk NY, USA) for Windows^®^/Apple Mac^®^. Syntax codes for all analyses are available upon request from the first author. Prior to inferential testing, data screening for violations of test assumptions occurred.

Though not directly related to the research aims, the authors first compared CA and MT scores between Professional and Amateur athletes to determine whether the responses could be grouped for the subsequent regression models. No differences in cognitive anxiety (*t*(159.21) = −0.45, *p* = 0.65, *d* = 0.03), somatic anxiety (*t*(162) = −0.45, *p* = 0.65, *d* = 0.01), self-confidence (*t*(162) = 0.4, *p* = 0.69, *d* = 0.03), or MT (*t*(169) = −0.65, *p* = 0.52, *d* = 0.07) were identified.

Three multiple linear regressions were then performed to investigate the extent to which Sportspersonship (opponent, rules, and game perspective) and MT predicted CA subscales (cognitive anxiety, somatic anxiety, and self-confidence). For all regression models, preliminary analyses were conducted to ensure no violation of the assumptions of linearity and homoscedasticity existed (determined via examination of variable relationship scatterplots and predicted residual scatterplots, respectively). All variables contained Tolerance scores above 0.1 and VIF scores below 10, indicating the absence of multicollinearity, in accordance with Tabachnick and Fidell [67]. Correlations amongst remaining predictor variables appear in Table 1. All correlations were weak to moderate, ranging between *r* = −0.04 and *r* = 49, indicating that multicollinearity was not present.

The first regression analysis examined predictors of cognitive anxiety. Preliminary tests indicated that Rules did not correlate with cognitive anxiety and was, therefore, removed from the analysis. The model was significant (*F*(3, 189) = 0.13; *p* < 0.001), explaining 13.4% of the variance. Participants with low MT (*β* = −0.3, *p* < 0.001) and greater respect for opponents (*β* = 0.23 *p* = 0.002) were more likely to exhibit cognitive anxiety (see Table 2).

Next, predictors of somatic anxiety were tested. The model was significant (*F* (4, 191) = 9.09; *p* < 0.001), explaining 16% of the variance. Similar to the cognitive anxiety model, the findings suggested that participants with low MT (*β* = −0.26, *p* < 0.001) and greater respect for opponents (*β* = 0.15 *p* = 0.05) were more likely to exhibit somatic anxiety (see Table 2).

The final model examined the predictors of self-confidence. Preliminary tests indicated that Rules did not correlate with self-confidence and were, therefore, removed from the analysis. This model was also significant, (*F*(3, 189) = 17.26; *p* < 0.001), explaining 21.5% of variance. The results suggested that mentally tough participants (*β* = 0.42, *p* < 0.001) displayed greater self-confidence during competitions (see Table 2). Post hoc power analyses using G*power 3.1.9.2 [72] suggested that all regression models were sufficiently powered to measure a medium effect (Cognitive Anxiety Model 1-β = 0.997, Somatic Anxiety Model 1-β = 0.995, and Self-Confidence Model 1-β = 0.997).

## 4. Discussion

Anxiety is a prevalent issue within sporting competitions and demands intensive investigation within combat sports due to its heightened effects on competition performance [73,74]. The present study examined the relationship between Mental Toughness (MT), Competitive Anxiety (CA), and Sportspersonship among CS athletes. The primary aim was to determine the extent to which MT predicted CA. As predicted, mentally tough CS athletes presented lower levels of cognitive and somatic anxiety, and higher levels of self-confidence in comparison to athletes with lower levels of MT. Similar relationships have been observed among non-combat sports athletes such as golfers [36] and handball players [61]. This relationship is consistent with Clough and colleagues’ [53] conceptual framework of MT which defines construct as a combination of commitment, confidence, control, and ability to positively appraise challenging situations. It is conceivable that athletes who possess confidence in their abilities, as well as a greater sense of control over both emotional and external outcomes, will be less affected by adversity during a competition. Thom et al.’s Goal-Expectancy-Self-Control (GES) model of MT [75] can also be applied to explain why mentally tough CS athletes experience less CA. The model posits that three psychological resources characterize the role of MT in fostering adversity in the face of competitive stress: goal setting, self-efficacy, and self-control. Inclinations toward setting specific and challenging goals are key components of MT [76,77], and one that allows athletes to improve their performance [78] and manage stress during adversity [79]. Thus, the model would suggest that mentally tough CS are more likely to set specific and challenging goals for both training and competitions that allow them to appraise competition stressors more positively. However, because goal-setting behaviors were not directly measured within the present survey, further evidence is needed to determine if goal setting is a key mediator of reduced CA. Self-efficacy is also a key component of MT [80]. Conversely, CS athletes with lower self-belief in their fighting abilities will be more likely to experience self-doubt and worry leading up to and during competition. Individuals with high self-efficacy show greater pain tolerance [81] and perseverance when faced with adversity [82], thus, CS athletes who feel efficacious are less likely to be mentally constrained by challenge-related stressors. The final component of the GES model, self-control is also a salient MT trait. Greater self-control in one’s emotions would allow one to modify their responses, including thoughts and emotions, to stressors [83]. As a result, mentally tough CS athletes would have a greater ability to manage how they react to competitive stress and more importantly, how they act in the face of these stressors.

A reduction in CA, and its adverse effects on performance [16,18,19] and commitment [38], would allow mentally tough CS athletes to compete with a better mindset and stay committed to their sport. MT can also provide benefits that go beyond reduced CA, such as adherence to rigorous weight-cutting regimes that are common within many CS. Athletes will often be required to restrict their diet and water intake, whilst increasing exercise to reach a required weight limit prior to competition [84,85]. In such contexts, an inherent drive to stay committed and in control of one’s actions is likely to be highly beneficial. Owing to these advantages, the current authors suggest that MT coaching could be an instrumental asset for CS success. Sports psychologists could look to foster MT among athletes as a means for reducing their CA. There is some evidence that psychological skills training can lead to an increase in self-reported MT [86,87]. Skills training programs promoting various sport-related behaviors such as visualization, relaxation, and thought stoppage have been successful in enhancing psychological qualities that underpin MT (i.e., hardiness, self-esteem, self-efficacy, dispositional optimism, and positive affectivity [88]). Goal-setting is another psychological skill coaches can promote to improve an athlete’s MT [55], evidence shows that it is a strategy routinely used by elite athletes to stay committed in psychologically and physically challenging contests [89]. However, the literature on MT coaching among adult athletes is currently limited. Field experiments are needed to determine the extent to which coaching can cultivate MT traits, the best approach for achieving this, and whether increasing an athlete’s MT will decrease their CA (i.e., via a repeated-measures design).

Other academics argue that MT development is a long-term process that is more dependent on earlier experiences at a younger age [90]. Though not relevant to adult CS athletes, the evidence does suggest that youth athletes can develop their MT through appropriate support from their parents and coaches. According to Crust and Clough [55], coaches can promote MT development by making training competitive and goal oriented, providing independent informational feedback (i.e., noting what was done right and what was done wrong), and encouraging reflective practice after competitive setbacks [91]—Thelwell and colleagues suggest that parents can play a similar role and have proposed specific educational programs for parents to cultivate MT.

The second aim of the study was to examine the relationship between CA and Sportspersonship. The researchers failed to replicate the observations of Kalkan and Yücel [43]; instead, findings suggested that individuals who held greater respect towards their opponents reported greater levels of somatic and cognitive anxiety. Though causal inferences cannot be made through cross-sectional data, the following explanations can be considered. One possible explanation for this observation may be that CS athletes with high levels of CA may be more empathetic towards their opponents due to a greater appreciation toward the negative stressors of the sport. Another possible explanation is that CA may be, in part, a result of reservations about having to inflict harm on another individual. As a result, athletes who hold less concern towards respecting their opponents may exhibit lower levels of CA emphasizing the complexities and nuances in sport. Alternatively, Sportspersonship and CA may be indirectly related through a mediating construct or a wider latent class, though further exploration into the individual differences of athletes is needed to explore this. Further investigation using a qualitative design would help to develop a more robust explanation for the relationship between respect towards opponents and CA. Indeed, the demonstration of Sportspersonship within CS can be seen as a good thing for the sport; however, due to a dearth of empirical knowledge within the current literature, it is difficult to ascertain whether such attitudes (or lack of) can be beneficial to competitive success.

Finally, although not pertinent to the main aims of the present study, the results suggest that professional and amateur CS athletes did not differ in relation to CA and MT. Whilst some studies have found similar null findings [50,92], others have demonstrated that experienced and professional athletes exhibit lower levels of CA [93,94,95] and higher levels of MT [96] than amateur athletes. The null findings suggest that within CS, MT—whilst advantageous—may not be pertinent to athletic longevity or progression to the professional level.

There were some limitations to the present study that need to be acknowledged. Firstly, the gender distribution of the sample was heavily skewed with only 7.37% of the sample being female. Despite CS being a male-dominated sport category—as reflected by the underrepresentation of female athletes in CS research [97,98,99,100]—the lack of responses from female CS athletes limits the generalizability of the findings. The study would have also benefited from a larger overall sample size to generate greater statistical power; however, given the small effect sizes that were observed for the null findings, it is unlikely that a larger sample will have changed the trajectory of the results. Furthermore, an inspection of the literature signifies that the current sample was adequately sized relative to existing CS studies (see [97]). As mentioned in the *methods*, in lieu of administrating the RCSAI-2 prior to competitions, the present study measured CA retrospectively. In doing so, the internal validity of the CA variables is questionable. In addition to evidence supporting the accuracy of retrospective anxiety reports [70], the current authors argue that even a retrospective measure of CA is of relevance to sports psychologists. Given that many of the adverse effects of CA occur through post-event reflection (e.g., sport discontinuation [38]), the effects of MT in reducing CA at any stage, whether it is prior, during, or retrospectively, is of great importance.

Despite these limitations, the current findings provide new academic insight into the relationship between MT and CA in CS, and highlight the potential value of MT coaching for fighters. The minimization of severe cognitive and somatic anxiety within CS may also help practitioners stay committed to the sport for longer by reducing negative experiences, though further academic inquiry into the pragmatic utility of MT coaching is needed. Future research should also explore the relationship between CA and illicit sporting behavior in CS, considering the fundamental influence such beliefs can have on unsporting behavior, such as the use of performance-enhancing drugs [101].

## Figures and Tables

**Table 1 behavsci-13-00713-t001:** Correlation coefficients for all continuous variables.

	1	2	3	4	5	6	7	8	9	10
1. Cognitive Anxiety										
2. Somatic Anxiety	0.56 ***									
3. Self-Confidence	−0.43 ***	−0.27 ***								
4. MT	−0.30 ***	−0.29 ***	0.44 ***							
5. Rules	0.12	0.23 **	0.02	−0.03						
6. Opponent	0.21 **	0.24 ***	−0.12 *	−0.04	0.33 ***					
7. Game perspective	0.12 *	0.28 ***	−0.24 ***	−0.28 ***	0.49 ***	0.44 ***				
8. Primary CS exp	0.02	0.01	0.10	0.07	−0.07	−0.03	−0.02			
9. Competition exp	0.07	0.01	0.06	0.02	−0.09	−0.02	−0.02	0.74 ***		
10. Total CS exp	0.13	0.10	0.04	0.02	−0.02	0.01	0.06	0.72 ***	0.77 ***	
Mean	25.52	22.56	30.75	34.04	3.17	3.53	2.99	8.10	8.1	11.15
Std Dev	8.32	7.44	7.44	5.52	0.62	0.52	0.69	5.78	6.77	7.95
Range	10–40	10–40	10–40	22–50	1.4–4	1–4	1–4	1–34	<1–40	1–46
Cronbach *a*	0.84	0.85	0.90	0.77	0.88	0.83	0.88	-	-	-
Skewness/Kurtosis	0.09/−0.95	0.48/−0.46	−0.48/−0.53	−0.01/−0.17	−0.22/−0.74	−0.7/−0.64	−0.79/0.77	1.5/2.81	1.71/3.81	1.48/2.83

* *p* < 0.05; ** *p* < 0.01; *** *p* < 0.001.

**Table 2 behavsci-13-00713-t002:** Regression model of cognitive anxiety subscales.

		Cognitive Anxiety					Somatic Anxiety					Self-Confidence			
	*R^2^*	*B*	*SE*	*β*	*t*	*R^2^*	*B*	*SE*	*β*	*t*	*R^2^*	*B*	*SE*	*β*	*t*
Model	0.13 ***					0.16 ***					0.21 ***				
Rules		—					1.59	0.94	0.13	1.69		—			
Opponent		3.68	1.2	0.23 **	3.06		2.13	1.07	0.15 *	1.99		−0.8	0.99	−0.06	−0.81
Game perspective		−0.78	0.94	−0.07	−0.83		0.81	0.91	0.08	0.89		−1.09	0.77	−0.11	−1.42
Mental Toughness		−0.46	0.11	−0.3 ***	−4.3		−0.35	0.09	−0.26 ***	−3.7		0.53	0.09	0.42 ***	6.06

Note. Statistical significance: * *p* < 0.05; ** *p* < 0.01; *** *p* < 0.001; — = variable removed from two models because it did not correlate with outcome variable.

## Data Availability

The data that support the findings of this study are openly available in the Open Science Framework at http://doi.org/10.17605/OSF.IO/XGCMN (accessed on 25 February 2022).

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
