# Peer review of "Competition Anxiety in Combat Sports and the Importance of Mental Toughness"

_behavsci, 2023, doi:10.3390/bs13090713_

Round 1

Reviewer 1 Report

First of all, I want to thank the Journal for the opportunity they have given me to review this article.

The article introduces a well-examined topic in sports science and sports psychology. However, it brings a novelty that makes it especially relevant. The use of MMA, where competitive anxiety and mental Toguhness is especially important. Congratulations to the authors for this thematic.

General Revision

In general, the authors have to make a thorough revision at the level of format and style.

For example, the citations, references and bibliography does not follow the guidelines of this journal. It is true that the magazine allows your references may be in any style, provided that you use the consistent format throughout. But the used style its not consistent. The citation style is very similar to the Apa rules, but the references do not follow that pattern, such as indentation. My recommendation is to follow the general MDPI regulations

Citations: In the text, reference numbers should be placed in square brackets [ ], and placed before the punctuation; for example [1], [1–3] or [1,3]. For embedded citations in the text with pagination, use both parentheses and brackets to indicate the reference number and page numbers; for example [5] (p. 10). or [6] (pp. 101–105).

References:

Books and Book Chapters:
1. Author 1, A.B.; Author 2, C.D. Title of the article. Abbreviated Journal Name Year, Volume, page range.

Books and Book Chapters:
2. Author 1, A.; Author 2, B. Book Title, 3rd ed.; Publisher: Publisher Location, Country, Year; pp. 154–196.
3. Author 1, A.; Author 2, B. Title of the chapter. In Book Title, 2nd ed.; Editor 1, A., Editor 2, B., Eds.; Publisher: Publisher 

Please review the regulations and fix the incorrect formats. Pay special attention to the year of publication, which should be in bold

https://www.mdpi.com/journal/behavsci/instructions

Abstract

This section has a maximum of 200 words of which the authors use 166 words. Perhaps the authors could spend a few more words commenting on the method on the study.

Introduction.- 

The introduction marks in a correct context the topics presented by the authors. It is wide and diverse. However, some aspects are missing:

Authors must incorporate more recent works in the introduction section. There are only three references above 2020 when the research is extensive today. They can consult works by authors such as Durand-Bush, Baker, among many others.

On the other hand, the authors talk about the Lazarus model on arousal, but omit the predominant IZOV model. This would help to explain some relationships that the authors present at this point. In addition, the authors could explain much better part of the results.

Material and Method.

The authors present the different instruments in a correct but incomplete way, assuming that the reader knows them. For example. Line 188 indicates that the psychometric properties are robust, but does not mention them. it would be convenient to briefly indicate the most relevant ones. Similar indications were produced with the rest of the instruments.

3.- Results. 

Line 254-256.- It would be convenient to show the results obtained.

Line 257.- in in accordance with. Tabachnick & Fidell, (2007)

Line 265.- Include in the footer of the table 2 all explanation of all the abbreviations of the same

Line 267.- Change variabl to variable.

Recommendation: The results do not present the power of the effect of multiple regression. I have calculated it with the free software G*Power and the results are very good. Authors should calculate and present it as it would provide strength to their results.

4.- Discussion.

I think the authors should restructure the discussion. The authors present a coherent discussion. However, I reiterate my recommendation to include more current cites and models on the matter, which will allow you to broaden the content of your discussion.

On the other hand, the objective of the work was to analyze the study variables in MMA fighters. It is not understandable that in all the discussion no reference was made to the type of athletes object of this work. We did not find any reference to MMA in the entire discussion, when the relevance of the work was to analyze it due to the peculiar characteristics of this sport. The results are discussed as if they were general athletes, and no explanations are given as to why the results may appear in MMA athletes.

Therefore the authors, taking advantage of the work they already have, should reconstruct the discussion based on the results of their athletes.

Author Response

We would like to thank the reviewer for their prompt and comprehensive review. We have addressed all of the reviewer’s comments and feel that the manuscript is much more robust thanks to the helpful feedback. We have marked all changed work within the manuscript using a blue font.

References:

Reviewer comment: the citations, references and bibliography does not follow the guidelines of this journal. It is true that the magazine allows your references may be in any style, provided that you use the consistent format throughout. But the used style its not consistent. The citation style is very similar to the Apa rules, but the references do not follow that pattern, such as indentation. My recommendation is to follow the general MDPI regulations

Authors’ response: We appreciate the reviewer’s recommendation, however, given that the journal permit researchers to use APA referencing formats and that the majority of the manuscript’s readers will be from psychological backgrounds, we have opted to stick with APA formatting. However, the reference list has now been properly formatted in accordance with the APA’s guidelines.

Abstract

Reviewer comment: This section has a maximum of 200 words of which the authors use 166 words. Perhaps the authors could spend a few more words commenting on the method on the study.

Author response: This section has now been expanded on to provide more details about the methodology, whilst still being within the 200-word limit.

Introduction.

Reviewer comment: Authors must incorporate more recent works in the introduction section. There are only three references above 2020 when the research is extensive today. They can consult works by authors such as Durand-Bush, Baker, among many others. On the other hand, the authors talk about the Lazarus model on arousal, but omit the predominant IZOV model. This would help to explain some relationships that the authors present at this point. In addition, the authors could explain much better part of the results.

Author response: We agree with the reviewer that some of the citations used in our literature review were dates (notably the papers that predated 2000). With that said, we feel that a requirement for research to have been published within the past 3 years is a little stringent, given the paucity of research on the relationship between mental toughness and cognitive anxiety. Moreover, research that has been published within the past decade is still considered modern and relevant by academics and this is seen within recent publications within this field – as can be seen in the meta-analysis of Fernandez et al (2020). However, we agree with the reviewers overall suggestion and have included a large number of moder citations within the literature review. As requested, we have also incorporated the Individual Zones of Optimal Functioning (IZOF; Hanin, 1997,2000) into our literature review. However, given that the model focusses on the impact of anxiety on performance, we have kept the discussion and application of this model brief. This is because our study did not look at performance and we would not like to detract away from the key focus of the paper.

Fernández, M. M., Brito, C. J., Miarka, B., & Díaz-de-Durana, A. L. (2020). Anxiety and emotional intelligence: comparisons between combat sports, gender and levels using the trait meta-mood scale and the inventory of situations and anxiety response. Frontiers in Psychology, 11, 130. https://doi.org/10.3389/fpsyg.2020.00130

Methods

Reviewer comment: The authors present the different instruments in a correct but incomplete way, assuming that the reader knows them. For example. Line 188 indicates that the psychometric properties are robust, but does not mention them. it would be convenient to briefly indicate the most relevant ones. Similar indications were produced with the rest of the instruments.

Author response: We initially removed the additional details because we wanted to keep the manuscript concise and followed the same writing protocols used in published articles within the field (see Dagnall et al., 2019; Papageorgiou et al., 2018). However, we have now included more details regarding the reliability indicators where the information is available (i.e., Internal consistency and Comparative fit indices).

Results

Reviewer comment: Line 254-256.- It would be convenient to show the results obtained.

Author response: We purposely did not include the statistical scores relating to all of our preliminary analyses as to not unnecessarily obfuscate the results section. Additionally, some of these analyses were determined through inspection of scatter plots. Our decision to do this is consistent with existing publications in psychology journals. However, we have provided more details to summarise these values (e.g., VIF and Tolerance scores) and also explain how the assumptions were confirmed.

Reviewer comment: Line 257.- in in accordance with. Tabachnick & Fidell, (2007)

Author response: All typos have now been addressed.

Reviewer comment: Line 265.- Include in the footer of the table 2 all explanation of all the abbreviations of the same

Author response: Table 2 variables are already presented in full terms and the meaning of each variable is explained within the methodology. If the reviewer is referring to the statistical symbols at the top of each column, these statistical symbols are universally accepted and we followed APA statistical guidelines by presenting them as they are. We hope that the reviewer is satisfied with our reasons for not adding more details to the footer.

Reviewer comment: Line 267.- Change variabl to variable.

Author response: All typos have now been addressed.

Reviewer comment: Recommendation: The results do not present the power of the effect of multiple regression. I have calculated it with the free software G*Power and the results are very good. Authors should calculate and present it as it would provide strength to their results

Author response: Post-hoc power analyses were conducted and are now reported.

Discussion

Reviewer comment: I think the authors should restructure the discussion. The authors present a coherent discussion. However, I reiterate my recommendation to include more current cites and models on the matter, which will allow you to broaden the content of your discussion.

Author response: The discussion has now be broadened to include more contemporary research findings. We have also incorporated a newer model to explain the association between Mental Toughness and Competitive Anxiety. We did not incorporate the IZOF model within the discussion because our findings did not look at the impact of anxiety and stressors on performance. In addition, when drawing out implications of the findings and recommendations from our results, we were cautious not to go beyond what the present evidence was able to reliably support, however, we have included further discussion about the MT-CA relationship.

Reviewer comment: On the other hand, the objective of the work was to analyze the study variables in MMA fighters. It is not understandable that in all the discussion no reference was made to the type of athletes object of this work. We did not find any reference to MMA in the entire discussion, when the relevance of the work was to analyze it due to the peculiar characteristics of this sport. The results are discussed as if they were general athletes, and no explanations are given as to why the results may appear in MMA athletes.

Therefore the authors, taking advantage of the work they already have, should reconstruct the discussion based on the results of their athletes.

Author response: The reviewer states that there was a lack of discussion directed specifically at combat sports, this is in part due to the scarcity of existing research looking at MT and combat sports. However, we have made a greater attempt to make the discussion more focussed on the anxiety and mental toughness of combat sport athletes.

Reviewer 2 Report

The work presents an interesting topic; it is a new topic, not in absolute terms but applied to combat sports.

- An important point has not been considered, and in my opinion, a paragraph should be added: the practice of cutting or making weight; is a well-established practice; it is estimated that practically all athletes have practiced it at least once in their career (10.3390/sports8100137 and 10.1123/ijsnem.2018-0165 for example).

Cutting weight sometimes consists of strong dehydration and/or strong caloric restrictions or even intense physical activities combined with heavy sweating, which has an important effect on the psychological sphere. Therefore a possible correlation and management in this sense should be shown.

- Schemes that can be applied for MT training would be interesting

it needs some revision

Author Response

Reviewer comment: The work presents an interesting topic; it is a new topic, not in absolute terms but applied to combat sports.

Author comment: We would like to thank the reviewer for their prompt and comprehensive review. We have addressed all of the reviewer’s comments and feel that the manuscript is much more robust thanks to the helpful feedback. We have marked all changed work within the manuscript using a blue font.

Reviewer comment: An important point has not been considered, and in my opinion, a paragraph should be added: the practice of cutting or making weight; is a well-established practice; it is estimated that practically all athletes have practiced it at least once in their career (10.3390/sports8100137 and 10.1123/ijsnem.2018-0165 for example). Cutting weight sometimes consists of strong dehydration and/or strong caloric restrictions or even intense physical activities combined with heavy sweating, which has an important effect on the psychological sphere. Therefore a possible correlation and management in this sense should be shown.

Author comment: thank you for the suggestion. We have included the studies and the issue of weight cutting to our discussion section, where we mention how MT could also be a beneficial trait for the difficult process of weight cutting. We have not discussed the effects of weight cutting within the literature review though for multiple reasons. Firstly, the competition anxiety measured in the present study was in relation to the anxiety felt by athletes during and right before their fights, therefore, any anxiety about weight cutting is not as relevant. Secondly, whilst there may be an argument that sever weight cutting could increase competition anxiety, there is currently no research evidence to support this. With the absence of direct evidence, we did not want to speculate within the literature review.

Reviewer comment: Schemes that can be applied for MT training would be interesting

Author comment: This is indeed an interesting application of the research and something that we are looking to explore in our future research. We have already provided some context around how MT has been fostered in other sports (within the discussion). We could have proposed further suggestions on how this could be done within combat sports, but to do so would be speculatory and based on opinion. We did not want to go beyond what our existing data (or previous findings) were able to evidence, therefore, we have kept the discussion of MT building schemes as they are within the discussion.

Reviewer comment: (English Language) needs some revision

Author comment: The full article has been reviewed , proof-read, and revised accordingly.

Round 2

Reviewer 1 Report

After the second revision I see that almost all the demands that I have raised in the first revision have been correctly solved for the most part. Where it has not been done, the reason has been justified.

I think the article is of sufficient quality to be accepted.